# Effect of pH on the structure and function of cyclin-dependent kinase 6

Mohd Yousuf[1☯], Anas Shamsi[2☯], Farah Anjum[3], Alaa Shafie[3], Asimul Islam[2], Qazi Mohd Rizwanul Haque[1], Abdelbaset Mohamed Elasbali[4], Dharmendra Kumar Yadav[5]*, Md. Imtaiyaz Hassan[2]*

1 Department of Biosciences, Jamia Millia Islamia, New Delhi, India, 2 Centre for Interdisciplinary Research in Basic Sciences, Jamia Millia Islamia, New Delhi, India, 3 Department of Clinical Laboratory Sciences, College of Applied Medical Sciences, Taif University, Taif, Saudi Arabia, 4 Clinical Laboratory Science, College of Applied Medical Sciences-Qurayyat, Jouf University, Sakaka, Saudi Arabia, 5 College of Pharmacy, Gachon University of Medicine and Science, Incheon City, South Korea

☯ These authors contributed equally to this work.
* dharmendra30oct@gmail.com (DKY); mihassan@jmi.ac.in (MIH)

**Data Availability Statement:** All relevant data are within the paper.

**Funding:** Funding Taif University, TURSP-2020/131, Dr. Farah Anjum Indian Council of Medical Research, ISRM/12(22)/2020, Dr. Md. Imtaiyaz

## Abstract

Cyclin-dependent kinase 6 (CDK6) is an important protein kinase that regulates cell growth, development, cell metabolism, inflammation, and apoptosis. Its overexpression is associated with reprogramming glucose metabolism through alternative pathways and apoptosis, which ultimately plays a significant role in cancer development. In the present study, we have investigated the structural and conformational changes in CDK6 at varying pH employing a multi-spectroscopic approach. Circular dichroism (CD) spectroscopy revealed at extremely acidic conditions (pH 2.0–4.0), the secondary structure of CDK6 got significantly disrupted, leading to aggregates formation. These aggregates were further characterized by employing Thioflavin T (ThT) fluorescence. No significant secondary structural changes were observed over the alkaline pH range (pH 7.0–11.0). Further, fluorescence and UV spectroscopy revealed that the tertiary structure of CDK6 was disrupted under extremely acidic conditions, with slight alteration occurring in mild acidic conditions. The tertiary structure remains intact over the entire alkaline range. Additionally, enzyme assay provided an insight into the functional aspect of CDK at varying pH; CDK6 activity was optimal in the pH range of 7.0–8.0. This study will provide a platform that provides newer insights into the pH-dependent dynamics and functional behavior of CDK6 in different CDK6 directed diseased conditions, viz. different types of cancers where changes in pH contribute to cancer development.

## Introduction

Cyclin-Dependent Kinase 6 (CDK6) is an important regulatory protein of the cell cycle and metabolism [1]. It controls the G1-S phase transition of the cell cycle through the Rb-E$_2$F pathway [2–5]. CDK6 is a key regulator in various cellular processes, such as cell proliferation [6–8], differentiation [9–11], inflammation [12], apoptosis and cancer [1, 8, 13, 14], suggesting

Hassan Indian Council of Medical Research, 45/54/ 2018-PHA/BMS/OL, Mohd Yousuf The funders had no role in study design, data collection and analysis, decision to publish, or preparation of the manuscript.

**Competing interests:** The authors have declared that no competing interests exist.

**Abbreviations:** CDK6, Cyclin-Dependent Kinase 6; ThT, Thioflavin T; EGF, Epidermal growth factor; CAK, Cyclin activating kinase; TNF-α, Tumor necrosis factor-α; PK, Protein Kinase.

the importance of CDK6 [15]. Its activity is directly activated by cyclin activating kinase (CAK) and cyclin D3, whereas indirectly several growth factors like tumor necrosis factor-α (TNF-α), vascular epithelial growth factor, epidermal growth factor(EGF), nerve growth factor, transforming growth factor-β (TGF-β) and cytokines play a role in its activation [15]. It is reported that CDK6 protein expression increases several folds than normal cell growth during tumor and tumor-associated diseases like cancer [16, 17]. In all cancers like colon, breast, lung, prostate, and stomach, the CDK6 expression level is very high, which signals cancerous cells to show resistance against the several drug molecules, chemotherapy and radiotherapy through continuous mutation [18].

In the last few decades, cancer cells show reprogramming of cellular and metabolic pathways. Several reports prove that the increased CDK6 level controls cell proliferation and alters the metabolic pathway of glucose consumption. CDK6 inhibits the Phosphofructokinase (PFK) and Protein Kinase (PK), key regulator enzymes of the glycolysis pathway. On the contrary, it activates the pentose phosphate pathway (PPP) and serine synthesis pathway that cause NADPH production and prevent ROS generation, which ultimately inhibits apoptosis [1, 19–23]. Evidence suggests that the alteration in the expression level of CDK6 involves cancer and is also found in neurodegenerative disorders like Parkinson's [16, 24–26]. All these reports highlight the importance of CDK6 as a potential therapeutic target to cure several pathological conditions [26], ranging from cancer to neurodegenerative disorders [27, 28].

There are additional domains in CDKs apart from consensus kinase domains. For instance, CDK16, CDK17 and CDK18 comprise a conserved catalytic domain flanked by amino and carboxy-terminal extensions implicated in the binding of cyclins. CDK12 and CDK13 comprise a kinase domain in the center with additional Arg/Ser rich motifs in the amino terminus that serve as docking sites for splicing factors assemblage and as splicing regulators.

CDKs belong to the CMGC group of kinases [29]. Cyclin-dependent kinases (CDKs) are serine/threonine protein kinase that controls the cell cycle [30]. CDKs and their cyclin strictly control every stage of the cell cycle by signaling checkpoints [31]. CDK6 gene is located on human chromosome 7q21.2, which encodes the 326 amino acids ~ 37 kDa cytosolic protein [32]. The human CDK6 structure is a single polypeptide chain that consists of N- terminal (amino-terminal) domain (5–100 amino acids) and C-terminal (Carboxyl terminal) domain (101–309). The N-terminal domain comprises five β strands with one PLSTIRE α- helix. C-terminal domain comprises mainly of α- helix with a small β sheet [33]. The all-inclusive structure of CDK6 is similar to CDK2 [34]. The catalytic site is present between both terminals. Cyclin-D interacts with amino-terminal and changes its structure, partially activating CDK6; CAK through phosphorylation/ dephosphorylation process at the specific site fully activated it [35, 36]. Several factors, including temperature, the strength of the buffer, pH, ions, macro and micro molecules present vicinity of the cell, is responsible for the function, proper folding and 3D structure of the protein [37–39]. Change in the ions concentration disturbs the pH homeostasis, leading to alteration of net charge on protein, ultimately affecting the protein function implicated in the diseased conditions [38]. It is apparent that any change in cellular pH, alters the cellular processes like cell growth, proliferation, and metabolic rate of cells; all these are hallmarks of cancer [40–42].

CDK6 structure and its function are already well reported, but structural and functional characterization at different physiological pH conditions is unknown. Every protein shows its optimum activity at a particular pH. Thus, this study intends to delineate the effect of various pH conditions on the structure and function of CDK6 protein. We have successfully cloned, expressed and purified the CDK6 protein from the bacterial system. The effect of pH on the secondary and tertiary structure of CDK6 was investigated employing multi-spectroscopic methods, fluorescence, UV visible, CD spectroscopies. ThT fluorescence was done to have an

insight to characterize the aggregate formation of CDK6. The structural studies were further complemented with the measurement of the kinase activity at various pH.

## Materials and methods

### Materials

DifcoTM LB broth Miller (Becton Dickinson, Fisher Scientific, Lenexa, KS, USA) was used for bacterial culture. Antibiotic Kanamycin (Sigma, Saint Louis, MO, USA) 50 μg/ ml was used. Isopropyl-β- thiogalactopyranoside (IPTG) was purchased from Calbiochem (USA). Tris-HCl, NaCl, NaOH, glycine, sodium acetate, sodium phosphate monobasic, dibasic, TritonX-100, dichloro-diphenyl-trichloroethane (DDT), lysozyme was procured from Sigma- Aldrich (St. Louis, USA). Ni- NTA column and beads purchased from BioRad (USA).

### Cloning, expression and purification

CDK6 protein was successfully cloned, expressed and purified using our well-established protocol as per earlier published literature [43].

### Sample preparation

CDK6 protein structure analysis was carried out using a wide range of pH 2.0 to 11.0 buffers. Glycine-HCl buffer was used for pH range 2.0–3.0; sodium acetate buffer was used for pH 4.0–6.0. Tris- HCl buffer was used for pH 7.0–8.0; sodium bicarbonate buffer was used for pH 9.0–10.0; Glycine-NaOH buffer was used for the pH range 11.0. The protein was incubated with a 50mM concentration of different buffers for 5 h at 25°C, and equilibrium was successfully attained before the spectroscopic measurements. All the measurements were carried out in triplicates form.

### CD measurements

Circular dichroism (CD) experiment was carried out at Jasco spectropolarimeter (J-1500, Japan) (PTC-517), equipped with a Peltier-type temperature controller to maintain the temperature. Far-UV CD spectra of CDK6 samples at different pH (2.0–11.0) were recorded in a 1 mm path-length cuvette with protein concentration kept at 8 μM. Spectra were recorded in the range of 205–250 nm with a 100 nm/min scan speed and a response time of 1 s [44]. Each spectrum was an average of 3 consecutive scans. The results were expressed as Mean Residue Ellipticity (MRE) in deg $cm^2$ $dmol^{-1}$ which is defined as:

$$MRE = \frac{MRWX\ \theta_{observed}}{10XlXc} \tag{1}$$

where $\theta_{observed}$ is the observed ellipticity in milli degrees, MRW is the mean residue weight of the protein, l being the path length in cm and c being the protein concentration in mg/ml.

### Fluorescence measurement

Fluorescence spectra of CDK6 were measured on Jasco spectrofluorometer (FP 6200, Japan) at 25°C ±1; the connected Peltier device-maintained temperature. The protein concentration was kept at 4 μM, and the sample was excited at 280 nm with the emission spectra recorded in the range of 300–400 nm [45]. Both the slit widths were fixed at 5nm and the response was set to medium intensity. All the measurements were carried out in triplicates and reported spectra were taken after subtracting appropriate blanks. For Thioflavin T (ThT) fluorescence, the sample was excited at 440 nm, and emission was recorded at 450–600 nm.

## UV/Vis absorbance measurements

UV/visible absorbance spectra of CDK6 were measured on Jasco UV/Vis spectrophotometer (V-660) connected with the Peltier device to maintain the temperature. The incubated sample spectra with a different range of pH buffer (2.0–11.0) were obtained at 240–340 nm by using a 1 cm path length cuvette. A Triplet set of samples were used for the experiment.

## Kinase activity assay

To study the effect of pH on the enzymatic activity of CDK6 protein, kinase assay was carried out in the presence of a different range of pH buffers. CDK6 protein (1 μM) sample was prepared in a different range of pH buffers and freshly prepared ATP was added (50 μM) in a reaction volume of 100 μl and incubated for 1hr at 25˚C for enzyme activity measurements. Similar reactions were carried out in triplet form for different ranges of pH buffers. To stop the reaction, a malachite green reagent (200 μl) was added into the reaction mixture and incubated for 30 minutes at 25˚C for the development of green color. The absorbance of the final product was measured spectrophotometrically at 620 nm.

# Results and discussion

The pH of the solution is a vital player in deciding the charge over the protein's surface, thereby influencing the chemical stability of the protein. Thus, it is apparent that the pH of the solution plays a key role and influences the structural and functional aspects of the proteins [46, 47]. Thus, considering the importance of pH in governing the protein's functionality, this study intends to delineate the effect of pH on the structure and activity of an important kinase, CDK6.

## Effect of pH on the secondary structure

To see the effect of pH on the secondary structure of CDK6, CD spectroscopy was deployed. CD spectroscopy is a sensitive technique routinely used to study changes in protein conformation under different pH conditions [48]. Native CDK6 showed spectral characteristics of a mixture of α-helix and β-sheet structures. The protein retained its native secondary structure in the pH range of 5.0–8.0, as revealed by the far-UV CD spectrum (**Fig 1A**). However, visible aggregates were observed in the pH range of 2.0–4.0 that interfered in CD spectra and hence almost no dichroic signal was obtained in this pH range. At pH 7.0–8.0, the protein attains a properly folded conformation evident from the far UV CD spectra obtained in this pH range.

**Fig 1B** shows far UV CD spectra of CDK6 in the alkaline range (pH 7.0–11.0). Far UV CD spectra are nearly identical in this range with no significant change suggesting that CDK 6

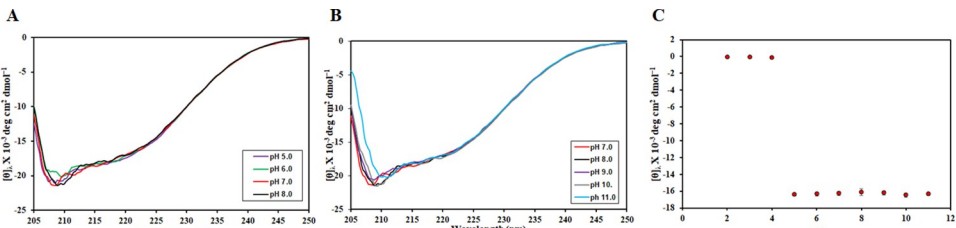

**Fig 1. Changes in the secondary structure of CDK6 measured by far-UV CD.** (A) Far-UV CD spectra of CDK6 in the pH range of 2.0–8.0 (B) Far UV CD spectra of CDK6 in alkaline pH range (7.0–11.0) (C) A plot of MRE at 222 nm ($\theta_{222}$) as a function of pH.

maintains a native-like conformation across the alkaline range with minimal alterations in the secondary structure over the entire range. **Fig 1C** shows the plot of $[\theta]_{222}$ versus pH and depicts that no significant perturbation of secondary structure occurs at alkaline pH (7.0–11.0). These observations suggest that CDK6 attains maximum structure, indicating a native-like structure at pH 7.0–8.0.

## Effect of pH on the tertiary structure

Intrinsic fluorescence studies were carried out to see the effect of pH on the tertiary structure of irisin. Since intrinsic fluorescence of proteins depends on aromatic amino acid residues [49, 50], a change in intrinsic fluorescence is a sign of change in the local environment of aromatic amino acid residues. Thus, any structural perturbation that affects the microenvironment around the fluorophore is often reflected in terms of changes in the emission spectrum of the protein [51]. CDK6 possesses 8 Tyr and 3 Trp residues, and hence, we have investigated the effect of pH on the tertiary structure of CDK6 in terms of intrinsic fluorescence spectroscopy. A characteristic redshift in the emission maxima is observed when a protein unfolds due to the increased solvent exposure of aromatic amino acid residues. Fluorescence emission spectra of CDK6 in the pH range (pH 2.0–8.0) are depicted in **Fig 2A**. CDK6 attains its native conformation in the pH range of 7.0–8.0, showing an emission maxima peak at 341 nm, as reported for other kinases [52]. A recently published study also reported that pyruvate dehydrogenase kinase 3 (PDK3) and Sphingosine kinase 1 (SPHK1) maintains their tertiary structure over the alkaline pH range [52, 53]. Another recently published literature reported that irisin, a myokine, maintains its structure in the alkaline pH range [39].

In the pH range of 5.0–6.0, there was no spectral shift in the emission maxima, suggestive of the fact that the environment around the aromatic amino acid residues was not perturbed to a significant extent. However, a decrease in fluorescence intensity was observed attributable to the protonation of water molecules or acidic amino acids surrounding Trp residues that act as dynamic quenchers of intrinsic fluorescence [54]. A spectral shift was obtained in the pH range of 2.0–4.0along with a decrease in the fluorescence intensity suggestive of the tertiary structural perturbations in this pH range. At pH 4.0, a decrease in the fluorescence intensity was coupled with a redshift of nearly 5nm, while at pH 2.0 and 3.0, visible aggregates were seen, and a blue shift of 3 nm and 6 nm was recorded, respectively. CDK6 maintains its native conformation at pH 5.0–8.0.

**Fig 2B** depicts the fluorescence emission spectra of CDK6 in the alkaline pH range (pH 7.0–11.0). It is quite clear from the figure that CDK6 maintains its tertiary structure in the alkaline range (pH 7.0 to 11.0), attributable to no significant change in $\lambda_{max}$ across the alkaline range. However, there was an evident decrease in the fluorescence intensity beyond pH 9.0.

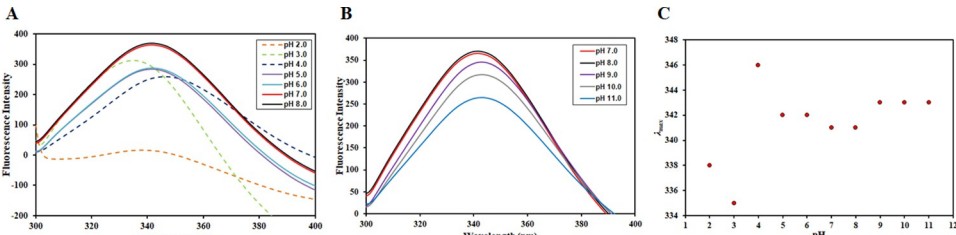

**Fig 2. Presenting pH-induced changes in the tertiary structure of CDK6 probed by fluorescence spectroscopy at 25˚C.** (A) Fluorescence emission spectra of CDK6 in the pH range of 2.0 to 8.0 (B) Fluorescence emission spectra of CDK6 in the pH range of 7.0–11.0. (C) Changes in emission $\lambda_{max}$ as a function of pH.

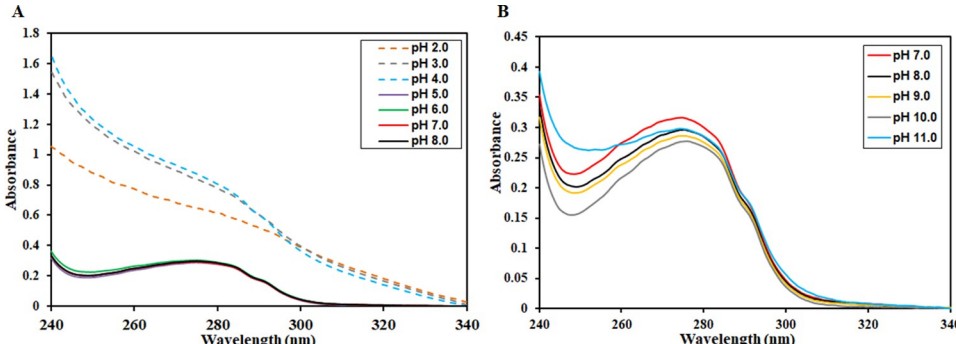

**Fig 3. Showing pH-induced changes in the tertiary structure of CDK6 were investigated by absorbance spectroscopy at 25°C.** (A) Absorption spectra of CDK6 in the pH range of 2.0 to 8.0. (B) Absorption spectra of CDK6 in the pH range of 7.0 to 11.0.

This decrease is attributable to the deprotonation of basic amino acids present in the vicinity of intrinsic fluorophores causing fluorescence quenching. Protonation/deprotonation of amino acid side chains may lead to charge destabilization in the local environment by disrupting the electrostatic interactions and internal salt bridges present in the native state [55]. The plot of $\lambda_{max}$ as a function of pH is depicted in **Fig 2C**; fluctuations were evident in $\lambda_{max}$ in the extreme acidic conditions, i.e., in the pH range of 2.0–4.0. In the pH range of 5.0–8.0, no significant change in the emission wavelength maxima was observed, suggesting no changes in the tertiary structure. However, a slight redshift of 2 nm was observed beyond pH 8.0. All these observations reveal that the tertiary structure of CDK6 is maintained in the alkaline pH range. In contrast, significant tertiary structural alterations occurred in the extreme acidic, with slight alterations occurring in the mild acidic conditions.

To further examine the effect of pH on the tertiary structure of CDK6, UV-vis spectroscopy was employed (240–340 nm). Conjugated double bond system present in the side chains of aromatic amino acids acts as a chromophore and absorbs light strongly in the UV region (240–340 nm) [56]. Additionally, the microenvironment of the aromatic residues corresponds to changes in UV absorption maxima; increased solvent exposure causes a blue shift in $\lambda_{max}$. **Fig 3A** shows the UV absorption spectra of CDK6 in the pH range (2.0–8.0). In the pH range of 5.0–8.0, nearly identical UV spectra were obtained, suggesting that minimal tertiary structural alterations occur and protein maintains its native-like conformation. In the pH range of 2.0–4.0, completely distorted UV spectra with a very high scattering were recorded. These distorted spectra suggest the distortion of the tertiary structure of CDK6 in the acidic conditions with subsequent aggregation. On the other hand, no significant changes were observed in the UV spectral profile of CDK6 across the alkaline range (pH 7.0–11.0), implying that CDK6 maintains its native-like conformation in the alkaline pH range.

These observations corroborate fluorescence and CD spectroscopy observations advocating that CDK6 maintains its native-like conformation in the alkaline pH range. Under extreme acidic conditions, distortion in its tertiary structure occurs with subsequent aggregation.

## Thioflavin T fluorescence

All the above observations suggest that aggregation of CDK6 occurs under extreme acidic conditions. Thus, ThT fluorescence was carried out to strengthen these observations further since ThT is an extrinsic dye routinely employed to characterize aggregates [57]. ThT binds to β-sheets, and in the case of aggregates, ThT-binding to aggregates causes an increase in the

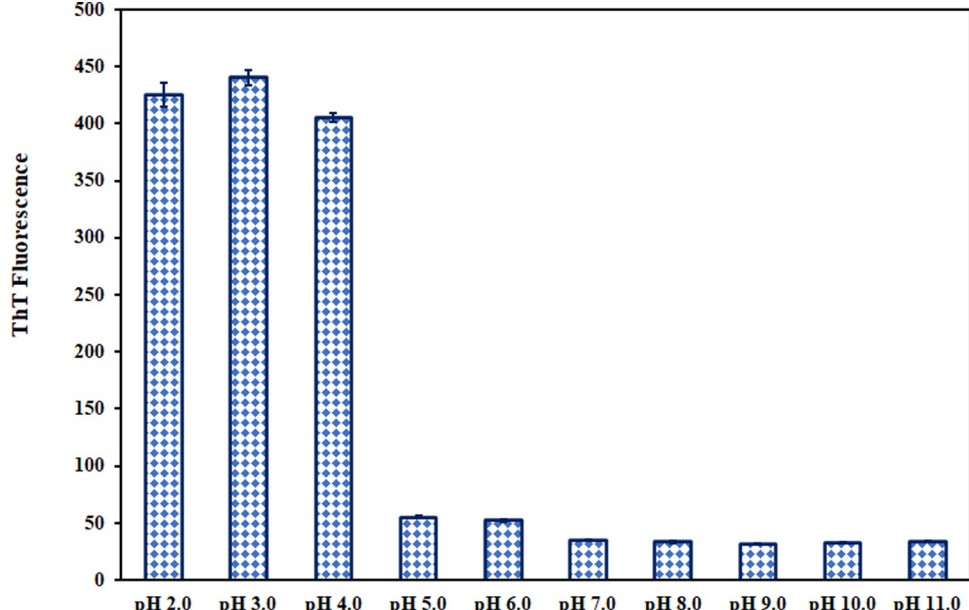

**Fig 4. Aggregates characterization by ThT fluorescence.** ThT fluorescence intensity was plotted as a function of pH (2.0–11.0).

fluorescence [58, 59]. **Fig 4** shows the ThT fluorescence intensity of CDK6 in the pH range of 2.0–11.0. It is apparent that aggregation occurs at pH 2.0–4.0, while no aggregation is observed at other pH levels. These observations are concurrent with other assays validating that CDK6 aggregation occurs in extremely acidic conditions.

## Effect of pH on kinase activity

The enzyme assay was carried out to delineate the effect of pH on the functional aspects of CDK6 as per previously published protocols [52, 60]. This assay involves the formation of a green-colored complex that is measured spectrophotometrically at 620 nm. CDK6 shows maximum enzymatic activity in the pH range of 7.0–8.0, suggestive of the fact that it exists in its native conformation. A significant decline in the enzymatic activity was observed moving away from this pH range in both directions (acidic and alkaline). **Fig 5** gives an overview of the enzymatic activity of CDK6 at different pH. The maximum activity of CDK6 at pH 8.0 was arbitrarily set as 100% to normalize the data. Our earlier spectroscopic observations revealed that CDK6 maintains its native-like conformation over the alkaline pH range. Thus, this loss in enzymatic activity in this alkaline pH range can be attributed to the ionization states of active site residues, which further affect catalytic activity. In the pH range of 2.0–4.0, CDK6 forms aggregates that interfere in this assay and thus, the enzymatic assay was not recorded at these pH conditions.

## Conclusions

It was discovered in this study that pH affects the structure and functional activity of CDK6. It was found that the pH of the solution influenced secondary and tertiary structural alterations in CDK6. CDK6 has a disrupted secondary structure when exposed to extreme acidic conditions (pH 2.0–4.0), resulting in the formation of aggregates in this pH range, according to our findings. Consistent with this finding, maximum ThT fluorescence was observed in the

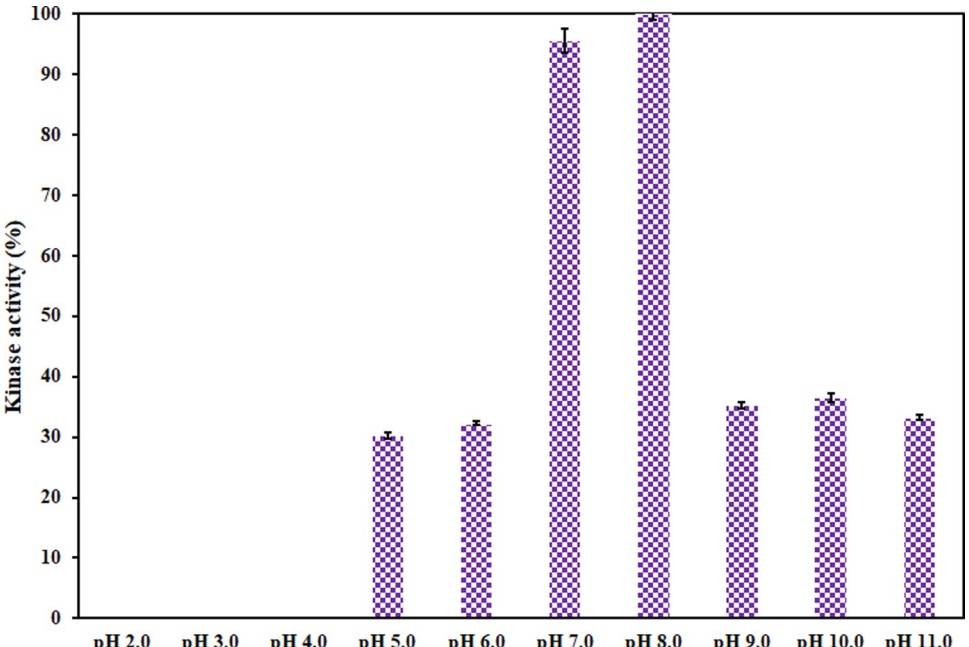

**Fig 5. The kinase activity of CDK6 was plotted as a function of pH (2.0–11.0).** The maximum activity of CDK6 was observed at pH 8.0, and it was arbitrarily set as 100% to normalize the data. Due to protein aggregation, the enzymatic activity was not determined in the pH range of 2.0–4.0.

extremely acidic conditions confirming the formation of CDK 6 aggregates under this condition. CDK6 maintains native-like conformation with no secondary structural alterations in the pH range of 5.0–8.0. CDK6 maintains its secondary structure over the entire alkaline range, evident from similar CD spectra.

Fluorescence and UV spectroscopic analysis revealed that CDK6 has a completely disrupted tertiary structure in the extremely acidic conditions. While slight alterations in the tertiary structure of CDK6 are observed in mild acidic conditions. CDK6 maintains its tertiary structure in the alkaline pH range. Enzyme assay revealed that CDK6 showed maximum kinase activity near physiological pH (pH 7.0–8.0), having an optimum pH value of 8.0. The change in the structure of CDK6 based on the pH can be further useful to understand the disease condition and cellular homeostasis to protein function under a variable range of pH conditions.

## Author Contributions

**Conceptualization:** Mohd Yousuf, Anas Shamsi, Dharmendra Kumar Yadav, Md. Imtaiyaz Hassan.

**Data curation:** Mohd Yousuf, Alaa Shafie, Asimul Islam, Qazi Mohd Rizwanul Haque, Md. Imtaiyaz Hassan.

**Formal analysis:** Anas Shamsi, Alaa Shafie, Abdelbaset Mohamed Elasbali, Md. Imtaiyaz Hassan.

**Funding acquisition:** Farah Anjum, Abdelbaset Mohamed Elasbali.

**Investigation:** Anas Shamsi, Farah Anjum, Abdelbaset Mohamed Elasbali.

**Methodology:** Anas Shamsi, Farah Anjum, Asimul Islam, Abdelbaset Mohamed Elasbali.

**Project administration:** Asimul Islam, Dharmendra Kumar Yadav.

**Resources:** Alaa Shafie, Qazi Mohd Rizwanul Haque, Dharmendra Kumar Yadav.

**Software:** Mohd Yousuf, Farah Anjum, Alaa Shafie, Qazi Mohd Rizwanul Haque, Dharmendra Kumar Yadav.

**Supervision:** Farah Anjum, Alaa Shafie, Qazi Mohd Rizwanul Haque, Dharmendra Kumar Yadav.

**Validation:** Mohd Yousuf, Asimul Islam, Qazi Mohd Rizwanul Haque, Md. Imtaiyaz Hassan.

**Visualization:** Mohd Yousuf, Asimul Islam, Qazi Mohd Rizwanul Haque, Md. Imtaiyaz Hassan.

**Writing – original draft:** Mohd Yousuf, Anas Shamsi, Md. Imtaiyaz Hassan.

**Writing – review & editing:** Asimul Islam, Qazi Mohd Rizwanul Haque, Abdelbaset Mohamed Elasbali, Dharmendra Kumar Yadav, Md. Imtaiyaz Hassan.

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
