## [Decision Letter · Decision Letter 0]

30 Dec 2021

PONE-D-21-38401Effect of pH on the structure and function of cyclin-dependent kinase 6PLOS ONE

Dear Dr. Hassan,

Thank you for submitting your manuscript to PLOS ONE. After careful consideration, we feel that it has merit but does not fully meet PLOS ONE’s publication criteria as it currently stands. Therefore, we invite you to submit a revised version of the manuscript that addresses the points raised during the review process.

We look forward to receiving your revised manuscript.

Kind regards,

Shailza Singh, Ph.D

Academic Editor

PLOS ONE

Journal Requirements:

Mohd Yousuf thanks the Indian Council of Medical Research for the award of Research Associateship. The authors sincerely thank the Department of Science and Technology, Government of India, for the FIST support (FIST program No. SR/FST/LSI-541/2012). 

Reviewers' comments:

Reviewer's Responses to Questions

**Comments to the Author**

1. Is the manuscript technically sound, and do the data support the conclusions?

Reviewer #1: Yes

Reviewer #2: Partly

Reviewer #3: Yes

2. Has the statistical analysis been performed appropriately and rigorously? 

Reviewer #1: N/A

Reviewer #2: No

Reviewer #3: Yes

3. Have the authors made all data underlying the findings in their manuscript fully available?

Reviewer #1: Yes

Reviewer #2: Yes

Reviewer #3: Yes

4. Is the manuscript presented in an intelligible fashion and written in standard English?

Reviewer #1: Yes

Reviewer #2: Yes

Reviewer #3: Yes

5. Review Comments to the Author

Reviewer #1: This study aims to understand the effect of various pH conditions on the structure and function of CDK6 protein. The authors have successfully cloned, expressed, and purified the CDK6 protein from the bacterial system in this study. Furthermore, the effect of pH on the secondary and tertiary structure of CDK6 was investigated employing a multi-spectroscopic approach viz. Fluorescence, UV visible, CD spectroscopy.

Overall, the manuscript was very well written. Moreover, the authors used various techniques and interpreted the results appropriately; therefore, the manuscript can be accepted with the following minor comments.

1. The authors can include some more recent references in the introduction section of the manuscript.

2. The authors can also include the 3D structure CDK in the introductory section and highlights the different functional domains.

3. The authors can also briefly include the future perspectives under the conclusion section.

4. The authors can provide the N and C terminus as Italics.

5. Some editing for the English language is required throughout the manuscript due to a few grammatical mistakes.

6. The authors can also include the CD in the abbreviation section.

Reviewer #2: The authors have studied structure function relationship of CDK6 at different pH conditions. I have some concerns and comments on the Figures mainly:

1. Figure 1 pH 4 shows no 0 absorbance in the CD spectra, the authors need to explain a bit more as to why this anomaly was observed? The experiments at each pH should be preferably done in triplicates and error bars for the data should be shown.

2. The Figure 1 is it raw data or was the CD spectra of the buffer subtracted to normalize it?

3. Since there is a drastic change between pH 4 and 5, the authors should definitely do two experiments at 4.4 and 4.8 pH to see how the curve changes, else there is no physical significance of this abrupt jump.

4. Figure 2A legends are missing only 3 pH value reported in fig what about the other pH values?

5. fluorescence intensity should be 300X10to the power something, it cannot be just 100.200 etc. Please remove negative fluorescence intensity from the plot. It makes no sense to show negative fluorescence data.

6. Was the fluorescence intensity normalized? If not then authors need to normalize their data.

7. Figure 3 Absorbance > 1 is a DIRECT VIOLATION of BEER LAMBERT's law. The authors should read on this and replot the UV data. Also the experiments should be done in triplicates for robust statistical analyses.

A general comment on Page 11 authors discuss perturbation in tertiary structure of CDK6. They specifically use the term "distortion". This is a generalized term. How did they measure distortion? What was the parameter that measured degree of distortion in the 3D structure is not clear at all.

Also they may use SEM images to visualize aggregates formed.

It is tough to base conclusions without adequate support.

I feel with these comments answered, the paper would be much more refined.

Reviewer #3: The paper "Effect of pH on the structure and function of cyclin-dependent kinase 6  " 

work's concept and approach are presented in a convincing manner. The study is adequate, and the PH experiments used are standard and well-established procedures. There are, however, a few comments and questions that must be addressed before the manuscript is accepted for publication.

1.  It is necessary to compare this to other kinases in order to determine whether the tertiary structure is preserved throughout the entire alkaline range.

2.   A significant structural perturbation occurs in extremely acidic conditions, whereas only minor structural alterations occur in mild acidic conditions, according to the findings. The particle size distribution of protein in acidic conditions can be validated by the author using Dynamic light scattering (DLS) technology.

3. The classification and numbering of headings and subheadings should be verified.

4. Thorough language editing is necessary to improve the presentation's quality. Numerous spacing errors must be corrected throughout the manuscript. In some instances, the use of prepositions is incorrect; please correct.

5. Abbreviations should be double-checked throughout the entire manuscript. They must first be defined before they can be used. As soon as the description is complete, the rest of the manuscript should be written entirely in abbreviations. Many sentences are unclear in places, such as the following: This study delineated the effect of pH on the structure and functional activity...................................subsequently forming aggregates in this pH range. It should be write It was discovered in this study that pH has an effect on the structure and functional activity of CDK6. It was discovered that the secondary and tertiary structural alterations in CDK6 were influenced by the pH of the solution. CDK6 has a disrupted secondary structure when exposed to extreme acidic conditions (pH 2.0-4.0), resulting in the formation of aggregates in this pH range, according to our findings.

6. PLOS authors have the option to publish the peer review history of their article (what does this mean?). If published, this will include your full peer review and any attached files.

Reviewer #1: No

Reviewer #2: No

Reviewer #3: No

---

## [Author Response · Author response to Decision Letter 0]

8 Jan 2022

Response to Reviewers' comments

Journal: PLOS ONE

Manuscript ID: PONE-D-21-38401

Title: Effect of pH on the structure and function of cyclin-dependent kinase 6

Reviewer #1

Reviewer #1: This study aims to understand the effect of various pH conditions on the structure and function of CDK6 protein. The authors have successfully cloned, expressed, and purified the CDK6 protein from the bacterial system in this study. Furthermore, the effect of pH on the secondary and tertiary structure of CDK6 was investigated employing a multi-spectroscopic approach viz. Fluorescence, UV visible, CD spectroscopy.

Overall, the manuscript was very well written. Moreover, the authors used various techniques and interpreted the results appropriately; therefore, the manuscript can be accepted with the following minor comments.

Response: Thanks for appreciating the scientific content and quality of the manuscript. We have addressed all the suggestions in the revised version of the manuscript. 

1. The authors can include some more recent references in the introduction section of the manuscript.

Response: Thanks!! We have updated the Introduction section citing latest references. 

2. The authors can also include the 3D structure CDK in the introductory section and highlights the different functional domains.

Response: We have updated the relevant section in the revised manuscript. 

3. The authors can also briefly include the future perspectives under the conclusion section.

Response: We have updated the relevant section in the revised version of the manuscript. 

4. The authors can provide the N and C terminus as Italics.

Response: Thank you for the suggestion. We have updated it in the revised version of the manuscript. 

5. Some editing for the English language is required throughout the manuscript due to a few grammatical mistakes.

Response: We have proofread the entire version of the manuscript using GRAMMARLY to remove all the grammatical errors. 

6. The authors can also include the CD in the abbreviation section.

Response: We have added the same in the abbreviations section. 

Reviewer #2: 

The authors have studied structure function relationship of CDK6 at different pH conditions. I have some concerns and comments on the Figures mainly:

Response: Thanks for reading and evaluating my work. We have addressed all the suggestions in the revised manuscript. 

1. Figure 1 pH 4 shows no 0 absorbance in the CD spectra, the authors need to explain a bit more as to why this anomaly was observed? The experiments at each pH should be preferably done in triplicates and error bars for the data should be shown.

Response: We agree with the reviewer. The experiments were carried out in triplicates and the error bars have been presented. We have updated the figures. No CD signal was shown because visible aggregates were present that interfered in CD spectrosocopy and hence no signal was obtained. To avoid any confusion, we have removed the CD spectra of pH 2.0-4.0 as no CD signal was obtained.

2. The Figure 1 is its raw data or was the CD spectra of the buffer subtracted to normalize it?

Response: All the reported spectra here are the subtracted spectra. 

3. Since there is a drastic change between pH 4 and 5, the authors should definitely do two experiments at 4.4 and 4.8 pH to see how the curve changes, else there is no physical significance of this abrupt jump.

Response: We observed aggregates in the range of pH 2.0-4.0 and hence, in this range we observed shifts coupled with changes in the intensity. Moreover, pH 5 is comparable to pH 7.0 and 8.0 as there is no shift obtained in this pH region. The abrupt behaviour of pH 4.0 is attribute to the presence of visible aggregates. 

4. Figure 2A legends are missing only 3 pH value reported in fig what about the other pH values?

Response: It was a formatting mistake and we have rectified. We have updated all the Figures.

5. fluorescence intensity should be 300X10to the power something, it cannot be just 100.200 etc. Please remove negative fluorescence intensity from the plot. It makes no sense to show negative fluorescence data.

Response: Thank you!! We have modified all the figures. The negative axis was shown as the pH 2.0 is showing fluorescence intensity in negative range due to presence of aggregates. 

6. Was the fluorescence intensity normalized? If not then authors need to normalize their data.

Response: All the spectra reported here are the subtracted spectra.

7. Figure 3 Absorbance > 1 is a DIRECT VIOLATION of BEER LAMBERT's law. The authors should read on this and replot the UV data. Also, the experiments should be done in triplicates for robust statistical analyses.

Response: We agree with the reviewer. This is a general phenomenon observed due to very high scattering associated with aggregates. We are aware of the fact that absorbance > 1 is not considered as per BEER LAMBERT law but this was just depicted to show the scattering occurring due to presence of aggregates and confirming the presence of aggregates. 

8. A general comment on Page 11 authors discuss perturbation in tertiary structure of CDK6. They specifically use the term "distortion". This is a generalized term. How did they measure distortion? What was the parameter that measured degree of distortion in the 3D structure is not clear at all. Also they may use SEM images to visualize aggregates formed.

It is tough to base conclusions without adequate support.

Response: Thank you for the suggestion. The term perturbation was used in respect to changes in secondary and tertiary structure observed from CD and fluorescence spectroscopy, respectively. No distortion was observed; we are taking only about structural alterations. 

Reviewer #3:

The paper "Effect of pH on the structure and function of cyclin-dependent kinase 6 " work's concept and approach are presented in a convincing manner. The study is adequate, and the PH experiments used are standard and well-established procedures. There are, however, a few comments and questions that must be addressed before the manuscript is accepted for publication.

Response: Thank you for the appreciation of the work and scientific merit of the study. We have addressed all the concerns in the revised version of the manuscript. 

1. It is necessary to compare this to other kinases in order to determine whether the tertiary structure is preserved throughout the entire alkaline range.

Response: Thank you for the suggestion. We have compared other kinases and updated the section in the revised manuscript. 

2. A significant structural perturbation occurs in extremely acidic conditions, whereas only minor structural alterations occur in mild acidic conditions, according to the findings. The particle size distribution of protein in acidic conditions can be validated by the author using Dynamic light scattering (DLS) technology.

Response: We agree with the reviewer that structural perturbation occurs in extremely acidic conditions, whereas only minor structural alterations occur in mild acidic conditions as proved by CD and fluorescence spectroscopy. 

3. The classification and numbering of headings and subheadings should be verified.

Response: Thank You!! The entire manuscript has been proofread to rectify all the mistakes. 

4. Thorough language editing is necessary to improve the presentation's quality. 

Numerous spacing errors must be corrected throughout the manuscript. In some instances, the use of prepositions is incorrect; please correct.

Response: We have used advanced version of GRAMMARLY to remove all the grammatical errors and typological errors. 

5. Abbreviations should be double-checked throughout the entire manuscript. They must first be defined before they can be used. As soon as the description is complete, the rest of the manuscript should be written entirely in abbreviations. Many sentences are unclear in places, such as the following: This study delineated the effect of pH on the structure and functional activity...................................subsequently forming aggregates in this pH range. It should be write It was discovered in this study that pH has an effect on the structure and functional activity of CDK6. It was discovered that the secondary and tertiary structural alterations in CDK6 were influenced by the pH of the solution. CDK6 has a disrupted secondary structure when exposed to extreme acidic conditions (pH 2.0-4.0), resulting in the formation of aggregates in this pH range, according to our findings.

Response: Thanks for the suggestion. We have proofread the manuscript and made the relevant changes as per reviewer’s suggestion. 

6. PLOS authors have the option to publish the peer review history of their article (what does this mean?). If published, this will include your full peer review and any attached files.

Response: Thank You. We don’t opt for that.

---

## [Decision Letter · Decision Letter 1]

25 Jan 2022

Effect of pH on the structure and function of cyclin-dependent kinase 6

PONE-D-21-38401R1

Dear Dr. Hassan,

We’re pleased to inform you that your manuscript has been judged scientifically suitable for publication and will be formally accepted for publication once it meets all outstanding technical requirements.

Kind regards,

Shailza Singh, Ph.D

Academic Editor

PLOS ONE

Additional Editor Comments (optional):

Reviewers' comments:

Reviewer's Responses to Questions

**Comments to the Author**

1. If the authors have adequately addressed your comments raised in a previous round of review and you feel that this manuscript is now acceptable for publication, you may indicate that here to bypass the “Comments to the Author” section, enter your conflict of interest statement in the “Confidential to Editor” section, and submit your "Accept" recommendation.

Reviewer #1: All comments have been addressed

Reviewer #2: All comments have been addressed

Reviewer #3: All comments have been addressed

2. Is the manuscript technically sound, and do the data support the conclusions?

Reviewer #1: Yes

Reviewer #2: Partly

Reviewer #3: Partly

3. Has the statistical analysis been performed appropriately and rigorously? 

Reviewer #1: N/A

Reviewer #2: N/A

Reviewer #3: Yes

4. Have the authors made all data underlying the findings in their manuscript fully available?

Reviewer #1: Yes

Reviewer #2: Yes

Reviewer #3: Yes

5. Is the manuscript presented in an intelligible fashion and written in standard English?

Reviewer #1: Yes

Reviewer #2: Yes

Reviewer #3: Yes

6. Review Comments to the Author

Reviewer #1: Authors addressed all the comments raised by the reviewers, therefore manuscript can be accpeted for publication.

Reviewer #2: (No Response)

Reviewer #3: All the  the comments are  addressed in a fair way. So, please let the article be published. All the  the comments are  addressed in a fair way. So, please let the article be published, and thank you for reading it.

7. PLOS authors have the option to publish the peer review history of their article (what does this mean?). If published, this will include your full peer review and any attached files.

Reviewer #1: No

Reviewer #2: No

Reviewer #3: No

---

## [Editor Report · Acceptance letter]

3 Feb 2022

PONE-D-21-38401R1 

Effect of pH on the structure and function of cyclin-dependent kinase 6 

Dear Dr. Hassan:

I'm pleased to inform you that your manuscript has been deemed suitable for publication in PLOS ONE. Congratulations! Your manuscript is now with our production department. 

Kind regards, 

on behalf of

Dr. Shailza Singh 

Academic Editor

PLOS ONE